# Demographic, Anthropometric and Food Behavior Data towards Healthy Eating in Romania

**DOI:** 10.3390/foods10030487

**Published:** 2021-02-24

**Authors:** Anca Bacârea, Vladimir Constantin Bacârea, Cristina Cînpeanu, Claudiu Teodorescu, Ana Gabriela Seni, Raquel P. F. Guiné, Monica Tarcea

**Affiliations:** 1Department of Pathophysiology, University of Medicine, Pharmacy, Science and Technology “George Emil Palade”, 540139 Targu Mures, Romania; anca.bacarea@umfst.ro; 2Department of Scientific Research Methodology, University of Medicine, Pharmacy, Science and Technology “George Emil Palade”, 540139 Targu Mures, Romania; 3Department of Community Nutrition and Food Safety, University of Medicine, Pharmacy, Science and Technology “George Emil Palade”, 540139 Targu Mures, Romania; cristina.cinpeanu@outlook.com (C.C.); claudiu.teodorescu4@gmail.com (C.T.); gabrielaseni@gmail.com (A.G.S.); monica.tarcea@umfst.ro (M.T.); 4CI&DETS/CERNAS Research Centres, Campus Politécnico, Polytechnic Institute of Viseu, 3500-606 Viseu, Portugal; raquelguine@esav.ipv.pt

**Keywords:** health, motivation, BMI, food behavior, education

## Abstract

Background: Each country has specific social, cultural, and economic characteristics regarding the motivations for improving health. The aim of this study was to evaluate demographic characteristics, anthropometric data, and elements related to food behavior and health, as well as Romanians’ motivations towards healthy eating. Methods: This is a descriptive cross-sectional questionnaire based study enrolling 751 Romanian participants, which was carried out in in 2017–2018. Results: We obtained a positive correlation between age and Body Mass Index, and this was maintained also when we analyzed the two genders separately, being, however, even stronger for women. The number of hours/day spent watching TV or in front of the computer was positively correlated with both age and BMI. In general, with aging, there is an increasing concern regarding the practice of a healthy diet. The higher education level was significantly associated with healthier choices. Conclusions: The study of the three dietary dimensions, food properties, health attitudes, and dietary behavior, vis-à-vis various disorders revealed that the group most concerned of their diet was those who suffered from cardiovascular disorders.

## 1. Introduction

There have been various studies regarding the impact of social and cultural factors upon different communities’ food behavior [1]. There is substantial evidence that social norms regarding food consumption strongly effect food choice, quality, and quantity consumed [2].

The globalization of agrifood systems has increased the availability and variety of foods through in food production and distribution changes. On one side, agricultural priorities rely on production and processing systems, markets, and livelihoods, with more concern for food safety and less care about general public health issues. Conversely, traditional public health focuses on agricultural issues that affect food security and the potential role of agriculture in preventing food-related diseases. We need to consider multidisciplinary aspects and the complex relationship between agribusiness, food consumption patterns, and health [3].

Adopting healthy diets can improve the nutritional behaviors and the status of population health. The guidelines released by the World Health Organization (WHO) establish a substantial reduction in the consumption of dairy products (by 28%), animal fats (by 30%), meat (13%), and sugar (by 24%) and a substantial increase in the consumption of cereals (by 31%), fruit (by 25%), and vegetables (21%) in order to reduce the burden of chronic diseases [4,5].

In the scientific literature, many variables have been used to analyze consumer behavior in the food market, the most frequent being the socio-demographic factors, motivations and attitudes, religious traits, and cultural and social background, along with geographical variability and lifestyle [6,7,8]. 

Over time, the Romanian people have undergone various lifestyle changes by adapting to food availability, information sources, and social status. In the early 1990s, the American model was adopted at the same time as the country’s modernization. Fast-food products emerged, and the number of supermarkets increased [9]. Highly processed foods with calorie-dense content, rich in carbohydrates, lipids, flavor enhancers, and food additives, have become popular among children and adolescents. Their popularity continues to grow insidiously due to the lack of the population’s targeted education [10]. Subsequently to the increase in obesity and the frequency of cardiovascular diseases, based on the development of nutrition-related mass-media projects, healthy food has been promoted in the last years. Consumers are urged to adopt a balanced diet to prevent food behavior disorders [11,12]. Many studies have focused on the importance of proper labelling (“low fat”, “low sugar”, “special price”) to support consumers’ choices [13,14]. A study shows that the probability of buying a product is higher when the price is low and the product is perceived healthier or tastier [15]. Another study found that price discounts seem to have ambiguous effects by promoting the purchase of healthy products, but also leading to increased calorie-dense purchases [16]. Unfortunately, most consumers are not aware or interested in reading food labels, depending on the social and cultural profile [17]. Many questions on what people understand about healthy eating still need to be answered.

Advertising is one of the major factors influencing the purchasing behavior of the population. Most of the TV advertising spots on food are dedicated to sweet, salty, and fat products (approximately 89%) [18]. Besides nutrients, people also check for food composition. An important role in choosing food products is played by the neo cortex, emotional eating, and education as well [19].

Worldwide, food behaviors are linked with the risk of occurrence of obesity [20], cardiovascular diseases [21], diabetes [22], respiratory disorders [23], psychiatric disorders [24], and cancer [25]. The role of maternal obesity on foetal development, birth outcomes, and child health is also recognized [26].

Romania, like many other countries, is making efforts to promote healthy eating. In order to do that, some particular aspects determining the motivations with regard to healthy eating should be considered and generated: like socio-demographic, cultural, economic, emotional, and environmental factors [27]. Despite the benefits of healthy eating, many people still prefer unhealthy food, and this indicates the need for more efficient community interventions.

The aim of this study was to evaluate demographic data, anthropometric data, and elements related to food behavior, as well as the Romanian people’s motivations towards healthy eating, as the first step for further development of health policies and strategies to improve nutritional behavior. To the best of our knowledge, this is the first Romanian study of its kind, and it is a part of a multinational project entitled “Psycho-social motivations associated with food choices and eating practices (EATMOT)” comprising 16 countries (Argentina, Brazil, Croatia, Cyprus, Egypt, Greece, Hungary, Italy, Latvia, Lithuania, Macedonia, Netherlands, Poland, Romania, Serbia, Slovenia, and The Unites States of America). 

This is a pilot study that aims to emphasize the importance of targeted education and community intervention, based on Romanian culture, attitudes, and motivations regarding healthy diets, and also aims to be a starting point for nutritional programs to be developed for Romanians all over the world. 

## 2. Materials and Methods 

This is a descriptive cross-sectional questionnaire-based study, targeted to evaluate the Romanian people’s motivations towards healthy eating, carried out during October 2017 and March 2018. The questionnaire was developed and validated within the EATMOT project by Ferrão et al. [6], and then it was translated into Romanian language. Our study includes 751 Romanian participants, from different regions of the country; thus, the study is country representative. The study was approved by the Ethics Committee of the University of Medicine and Pharmacy, Science, and Technology “G.E. Palade” Targu Mures and was conducted in accordance with the Helsinki Declaration.

We included only adult people, aged 18 and older, who fully completed the whole questionnaire. Participants had to answer two sets of questions: the first set of questions referred to demographic data, anthropometric data, and elements related to behavior and health, and the second set referred to motivations for health. The analyzed parameters in the first set of questions were age (we divided the population into five age categories: 18–29, 30–39, 40–49, 50–59, and ≥60 years old), gender, weight, height, environment (urban, suburban, rural), the last level of studies completed (general school, high school, college), marital status (single, married/living together, divorced/separated, widow), current employee status (employed, unemployed, retired, working student), field of activity/specialization in certain areas (nutrition, food, agriculture, sports, psychology, health-related activities), responsibility for eating, physical activity (never, sporadic, occasionally, moderate, intense), hours/day spent in front of the TV or computer, own opinion about having a healthy/balanced diet (never, rarely, sometimes, frequently, always), and the presence of chronic disorders (cardiovascular disease, diabetes mellitus, high cholesterol, high blood pressure, gastric disorders, intestinal disorders, obesity, or others). We calculated the BMI in kilograms divided by the square of height in meters (kg/m^2^) based on the declared weight and height. We looked for significant associations and differences among age, BMI, and the studied parameters.

The second set of questions comprised ten items, as follows:Q1—I am very concerned about the hygiene and safety of the food I eatQ2—It is important for me that my diet is low in fatQ3—Usually, I follow a healthy and balanced dietQ4—It is important for me that my daily diet contains a lot of vitamins and mineralsQ5—I do not avoid foods, even if they may raise my cholesterolQ6—I try to eat foods that do not contain additivesQ7—I do not eat processed foods, because of their lower nutritional qualityQ8—It is important for me to eat food that keeps me healthyQ9—I do not avoid foods, even if they may raise my blood glycaemiaQ10—I avoid foods with genetically modified organisms

The items included in the second set to answer all ten questions regarding health motivations were: totally disagree, disagree, neither agree nor disagree, agree, or totally agree. The ten questions investigated the participants’ interest in food composition (Q1, Q6, and Q10) and healthy properties (Q2, Q3, Q4, Q5, Q7, Q8, and Q9), and together they aimed to show the general picture regarding Romanians’ motivations for a healthy diet. Q3 is a frequency-based question, whereas the others are attitudinal questions. Q5 and Q9 were reversed, the question referring to the healthiest attitude being on the left (disagree), compared to the other questions, where the healthiest attitude was on the right (agree). Because these items were measuring different things, we opted to create two composite scales, one for food properties (Q1, Q6, and Q10) and one for health attitudes and motivations (Q2, Q4, Q5, Q7, Q8, and Q9), leaving Q3, referring to healthy diet frequency, to stand alone. We decided to group the questions in this manner, based on face validity and based on the results of the previous studies [28]. In order to do the statistical analyses, we first reversed Q5 and Q9.

We used Microsoft Excel for Mac 2011 (Microsoft Corporation, Redmond, WA, the USA) for data collection and handling, and Graph Pad Prism demo version (Graph Pad Software, La Jolla, CA, the USA) and Epi Info version 7 (Centers for Disease Control and Prevention, Atlanta, GA, USA) and SPSS Statistics v.25 for statistical analyses. We used statistical methods to provide mean and SD for continuous variables or median and range for discrete variables, and absolute and relative frequency counts for categorical variables. The Student *T*-test and Mann–Whitney U test were used as appropriate statistical tests to compare continuous variables between the groups (normal or non-Gaussian distribution); for correlations, we used the Pearson or the Spearman test according to variables distribution. To establish a mean difference between several continuous variables, we used the ANOVA test for Gaussian distributions and the Kruskal–Wallis test for non-Gaussian distributions [29]. A *p*-value under 0.05 was considered statistically significant. The item analysis was performed using the Pearson correlation coefficients, and the associations were interpreted as not existing (r = 0), very weak (0.00 < r < 0.10), weak (0.10 ≤ r < 0.30), moderate (0.30 ≤ r < 0.50), strong (0.50 ≤ r < 0.70), very strong (0.70 ≤ r < 1), or perfect (r = 1), according to the value of r [6]. The internal consistency of the scales was evaluated by using Cronbach’s alpha, according to Marôco [30], as follows: over 0.9: excellent, 0.8–0.9: very good, 0.7–0.8: good, 0.6–0.7: medium, 0.5–0.6: reasonable, below 0.5: bad. 

## 3. Results

The characteristics of the studied population: socio-demographic data, environment, professional areas, physical activity, and medical history can be found in Table 1.

The BMI values were calculated for the whole sample and varied between 15.05 and 43.57 kg/m^2^, being on average 24.59 ± 4.34 kg/m^2^. In Table 2, we evaluated our subjects’ BMI and age in relation to the studied parameters.

We obtained a positive correlation between age and BMI, and this was also maintained when we analyzed the two genders separately, this correlation being stronger in women. When analyzing the marital status, we obtained statistically significant differences between single vs. married (*p* < 0.0001) and between single vs. divorced (*p* = 0.0382), but not between single vs. widow (*p* = 0.2386). A continuation of this study to include a higher number of subjects in the widowed category is needed to confirm this result. This was also the case of agricultural worker as a subcategory of professional activity, for which we obtained significantly higher BMI, but the number of cases for this category was lower.

Regarding the number of hours/day spent r in front of the TV or computer, we obtained positive correlations for both age and BMI. 

We obtained no significant associations among BMI, environment, current professional activity, responsibility for eating, and physical activity.

We displayed in Table 3 the results of participants’ answers to questions regarding motivations towards healthy eating.

Except for Q2, where the highest percentage was obtained for the item “neither agree nor disagree”, for the other questions, the highest percentage was registered for items “agree” and “totally agree”.

Table 4 shows item–item correlations for the group of questions investigating participants’ attitudes toward food properties. The results indicate moderate correlations. The value of Cronbach alpha was 0.689, which is a medium value, based on which we accepted all three questions in composite scale for food properties.

Table 5 shows item-item correlations for the group of questions investigating health attitudes and motivations. Because of the negative, very weak, and weak associations between the reversed Q5 and Q2, Q4, and Q7, between Q7 and Q4 and Q5R, and between the reversed Q9 and Q2, Q4, Q5R, Q7, and Q8, we considered eliminating Q5, Q7, and Q9. Q5 and Q9 are negative questions, and the answers were probably inconsistent due to the participants’ lack of attention.

When eliminating these three questions, the Cronbach alpha increased to 0.712, which was interpreted as a good value. 

The associations between composite scale for food properties, refined scale for health attitudes, reported dietary behavior (Q3), and investigated variables (*p* values) are shown in Table 6.

With aging, there was an increasing concern for a healthy and balanced diet, but this was not reflected when analyzing food properties in relation to age, perhaps because of lack of knowledge, especially for Q10. However, there was a positive correlation between aging and the three studied dimensions (for food properties and aging r = 0.1408, *p* = 0.0001; for health attitudes and aging r = 0.1643, *p* = 0.0001; for dietary behavior and aging r = 0.1490, *p* = 0.0001). The age category of 18–29 was the most interested in food properties, although this was statistically not significant. The age category ≥ 60 was the most concerned age group about their health and a healthy and balanced diet compared to the other four age groups. The low number of subjects included in the age category ≥ 60 (N = 44) was a limitation of our study, and our results should be confirmed by including a larger population in the study. Women answered more with “agree” and “totally agree” to health attitudes and dietary behavior (Q3) questions than men. Regarding food properties, there was no difference between men and women.

We obtained significantly more “agree” and “totally agree” answers for item Q3 in the case of participants who came from urban environments. For composite scale investigating food properties and health attitudes, respondents from urban environments answered more with “agree” and “totally agree”, even if this was statistically not significant.

College education level was significantly associated with health motivations for items investigating health attitudes (college education 52.84% vs. high school 39.52%) and dietary behavior (college education 48.02% vs. high school 31.14%). Even if this was statistically not significant, persons with a college education were more preoccupied with food properties than the persons belonging to the other educational groups.

There were no correlations between BMI, food properties, and attitudes toward health. For Q3, there was a weak negative correlation with BMI. 

Most of the respondents practicing occasional and moderate physical activity answered agree or totally agree for all three investigated dimensions.

TV/computer hours were not correlated with food properties, healthy attitudes (very weak correlation r = 0.0901), or dietary behavior. 

People with cardiovascular disorders were more often preoccupied with a healthy diet (92.59% of those who reported cardiovascular disorders answered “agree” and “totally agree” to the questions investigating health attitude). Although statistically not significant, people with cardiovascular disorders answered more often “agree” and “totally agree” for both composite scales: food properties and dietary behavior (Q3). Of those who reported cardiovascular diseases, 7.41% also reported hypercholesterolemia.

People reporting gastric disorders were preoccupied with all three studied dimensions, and those having intestinal disorders were especially concerned about food properties. This finding is not maintained for composite scales investigating health attitudes and dietary behavior, where the individuals without intestinal disorders are more concerned about healthier choices, although statistically not significant.

Persons who reported obesity show a lack of interest regarding all three investigated dimensions, which was significant for reported dietary behavior and not significant for food properties or heath attitudes. We did not interpret the category including other disorders because of the low number of subjects in this group.

We found a significant relationship (*p* = 0.0000) between reported dietary behavior and health attitudes: 224 subjects answered “agree” and 116 subjects answered “totally agree” for both composite scale investigating health attitudes and for Q3, investigating dietary behavior. Analyzing the different age groups, we obtained the following percentages, showing the concordance in responses “agree” and “totally agree” for these two dimensions: 36.53% for the age category 18–29 years old, 46.87% for the age category 30–39 years old, 50.80% for the age category 40–49 years old, 48.48% for the age category 50–59 years old, and 59.09% for the age category ≥60 years old. In other words, with aging, participants’ answers are consistent with healthier choices.

## 4. Discussion

Modern lifestyle induces harmful behavior regarding eating and physical activity [31]. Altered behaviors are a growing problem in Romania, such as in other countries. Obesity is one important disease associated with unhealthy eating behaviors, and the fact that Romania is a middle-income country can contribute to the obesity epidemic spreading [32]. As our results indicated, obesity was the most frequent health problem reported by the participants in this study (6.66%). Since we have already shown a significant positive association between BMI and glycaemia in the age category older than 22 [33], we consider it appropriate to evaluate the BMI in relation to demographic, anthropometric data, and elements related to behavior and health. Similar to the results of Abdella et al. [34], age positively correlated to BMI. We also demonstrated a positive correlation among BMI, age and the number of hours spent in front of the TV or computer. In their study, Martínez-Moyá et al. [35] proved that the number of hours spent watching television and lower physical activity were significantly associated with a higher BMI in young adults [35]. Other studies showed that an increased screen time spent was significantly associated with the risk of obesity, but not the physical activity level [36], and watching television is the leading sedentary activity in association with obesity [37]. This relation was also found in children; according to Golshevsky et al. [38], higher BMI was associated with more hours spent watching television, and less time spent in organized sports activities. The increased number of hours spent in front of the TV with aging can be related to life cycle changes. When we separately analyzed the relationships between the two genders, the correlation was stronger for women, possibly because of the hormonal changes related to aging. When comparing the BMI between the two genders, women had lower mean BMI values than men (23.97 vs. 25.91), possibly because women’s mean age was lower than men’s (36.6 vs. 41.04).

Many studies emphasized the role of education in BMI control [39,40]. However, we obtained higher BMI values for the group with college education compared to the group with high school. Some explanations for this finding could be easy access to college education and the lack of physical activity. On the other hand, the group with a college education has the highest mean age, so the BMI can be attributed to aging. Strategies to improve knowledge about healthy eating must be developed to have a better weight control and focus on different age categories.

There was no correlation between the environment and BMI, and this is quite normal considering the people’s migration and easy access to information regardless of the native environment.

The marital status also influenced the BMI, married persons having higher BMI values compared to single persons. This is in concordance with the results of other studies [41,42]. It is unclear how marital status affects BMI, probably by changing the body weight-related perceptions and eating behaviors [42]. Some studies indicated that increased BMI affects the status of the employee because of health problems or because it decreases the chances to be employed [43,44]. We did not find BMI differences between employed and unemployed participants, probably because in Romania many young adults prefer not to work because of low income, and they benefit from social assistance (33.41 vs. 41.39 years).

Normally, professional activity is associated with different knowledge regarding health and with different physical activity level. The highest mean BMI value was obtained for those working in agriculture, but the result is debatable due to the small number of subjects in this group.

Even if we did not find a statistical difference for BMI between the different categories of physical activity, we must notice that the mean age of subjects performing intense and moderate physical activity is lower than in those with no or sporadic physical activity. This is a good aspect, showing the concern of younger adults for their health. Many studies showed the benefits of physical activity upon health, associated with a decreased risk of cardiovascular events [45] and better control of blood glucose level [46]. Carraça et al. [47], in their recent study, identified a behavioral pattern showing that adults who are not interested in physical activity are women, have a higher BMI, have been less educated, and are unemployed. Their eating habits are more likely to be less healthy, and they perceive more barriers when it comes to physical activity [47]. Another study shows that barriers to healthy eating and/or physical activity significantly influenced BMI, the level of physical activity, stress, and fruit and vegetable intake [48].

Consumers’ beliefs and knowledge about healthy foods are variable. A food is considered healthy in general if it is low in total fats and saturated fats, and meets certain requirements regarding cholesterol and certain vitamins or minerals content [2,49,50]. In his study, Lusk identifies four categories on how healthy food should be defined: based on food nutrients, the entire composition of the food, nutrients from the whole diet, and based on holistic consumption patterns, and the respondents were almost equally distributed among these categories [28].

The fact that a person has adequate knowledge and answers the questions accordingly, does not always mean that this knowledge is applied in everyday life. In a study, the participants showed adequate nutrition knowledge, but eating behavior was strongly influenced by social and physical environmental factors [51]. Mete et al. [52] underline the role of social media in improving healthy food choices by promoting healthy eating information.

This is the motivation for our analysis of the questions in three dimensions—food properties, health attitudes, and separately the general question for dietary behavior (Q3). Looking at food properties (hygiene, additives, and genetically modified organisms) in relationship with the studied parameters, we can see that more physically active people and those having gastric and intestinal disorders were more preoccupied with these parameters. The associations were also statistically significant between physical activity, health attitudes, and dietary behavior (Q3), indicating that these people were concerned and motivated to maintain their health. Genetically modified food is a controversial subject and involves important knowledge [53]. People with gastric and intestinal disorders relate food properties to their disease. We assume that this is why they were more preoccupied with healthy diets compared with other people suffering from other pathologies.

With aging, people were more preoccupied with making healthier choices (significant associations between age and refined composite scales regarding health attitudes and Q3), probably because of changes related to aging and occurring pathologies. This is similar to the results of Whitelock et al. [54], where the participants described efforts focusing on avoiding foods high in fats and sugar content. More women than men had perceptions compliant with a healthy diet, possibly because they have better nutritional knowledge and interest for it [27,55,56].

We obtained significantly more answers compliant with a healthy and balanced diet in general for urban areas. However, this difference was not maintained concerning food properties and health attitudes. This can be due to an increased general interest in a healthy diet that is not translated into specific choices. Education level is essential in connection with BMI and various metabolic diseases [40,57]. As expected, we obtained more correct answers in the college group regarding healthy diet and fat, vitamins, and minerals content, but these answers were not reflected in the case of a lower BMI. 

In one study about the perception of healthy eating in Romania, it was shown that tradition is very much related to eating behavior, and was correlated with BMI [31]. Lotrean et al. [58] performed a study among Romanian students and revealed three main dietary structures: two of them protective against becoming overweight, but different regarding physical activity, and the third one (fast food diet) associated with higher BMI and lack of daily physical activity [58]. We believe that people have somewhat imbalanced attitudes about food and healthy eating, which could significantly affect the transposition of beliefs, knowledge about healthy eating, and attitudes into behavior. The study subjects having different disorders showed lack of interest regarding healthy eating choices, which can be a contributing key factor to the evolution of their diseases.

According to Nagata et al., after a seven-year follow-up of young adults with overweight/obesity and unhealthy weight control behaviors at baseline, they still had higher BMI than those without unhealthy weight control behaviors [59].

We found a significant association between health attitudes and cardiovascular disorders, which is logical. People with cardiovascular diseases are more preoccupied to have a particularly low-fat diet.

Traditionally, recommendations were made for individual nutrients consumption such as saturated fats, sugar, sodium, and cholesterol in the diet, because they are usually over-consumed by many people and are linked to the development of chronic diseases. These can also lead to erroneous effects [60,61,62]. One controversial topic was the association between saturated fats and cardiovascular diseases without considering substitute nutrients and cholesterol, and another one is the association between cholesterol and cardiovascular diseases, which is confused with the intake of saturated fats [63]. The contemporary dietary guide recommends healthy dietary models, with an emphasis on food-based recommendations. The diet as a whole, meaning the combinations and the amounts of food (nutrients) we eat daily, is an essential determinant of health [63].

Our study has some limitations because of gender differences (more women than men) and low number of participants in some categories (e.g., agriculture as a field of activity, age group ≥ 60 years old). Additionally, the participants were not from all counties of the country, so the study is not 100% country representative. We calculated the BMI based on the self-reported values for weight and height, so some bias occurred.

We showed that with aging, people were more preoccupied with making healthier choices. However, it is not only the occurrence of various diseases that should make people aware of healthier choices. Hence, there is a need for an intensive national strategy for health motivations. According to age groups, this strategy should be addressed differently, knowing that radical changes in lifestyle are difficult to accept with age. Another important aspect is the translation of information about healthy choices into real choices. According to our findings, obesity was the most frequent health problem reported by the participants in this study, and despite our expectations, we obtained higher BMI values for the college education group, although they chose the correct answers for their health when asked. This indicates the need for a long-term strategy to motivate people to make healthy eating choices, starting with children and involving also their families [58].

## 5. Conclusions

We obtained a positive correlation between demographic parameters and the BMI in the Romanian population; also their healthy food behaviors were stronger for women. The number of hours/day spent watching TV or in front of the computer was positively correlated with age and BMI. The higher education level was significantly associated with healthier choices regarding nutrition practices and motivations. Regarding the associations between the sociodemographic characteristics and different disorders, we observed that the subjects with cardiovascular disorders were more preoccupied with healthier diets in most cases.

Nutritionists, specialists in medicine, and food stakeholders should promote healthy diets through adequate sources of information aimed at target groups. They should develop a more efficient strategy to motivate people to make healthy eating choices and improve Romanian food behavior. 

## Figures and Tables

**Table 1 foods-10-00487-t001:** Demographic, anthropometric data, and elements related to food behavior and health status of the studied population.

Parameter	N ^(1)^	(%)
Age (years)	751	100%
18–29	260	34.62%
30–39	128	17.04%
40–49	187	24.90%
50–59	132	17.58%
≥60	44	5.86%
Gender		
Female	511	68.04%
Male	240	31.96%
Education		
General school	3	0.40%
High school	167	22.24%
College	581	77.36%
Environment		
Urban	623	82.96%
Suburban	26	3.46%
Rural	102	13.52%
Marital status		
Single	211	28.10%
Married/living together	480	63.91%
Divorced/separated	50	6.66%
Widow	10	1.33%
Employee status		
Employed	542	72.17%
Unemployed	31	4.13%
Retired	29	3.86%
Working student	149	19.84%
Professional area		
Nutrition	97	12.92%
Food	48	6.39%
Agriculture	13	1.73%
Sports	33	4.39%
Psychology	35	4.66%
Health-related activities	332	44.21%
Professional activity is not related to any of the above areas	269	35.82%
You are responsible for what you eat		
Yes	699	93.08%
No	52	6.92%
Physical activity		
Never	52	6.92%
Sporadic (<1 time/week)	253	33.69%
Occasionally (1 time/week)	229	30.49%
Moderate (2–3 times/week)	170	22.64
Intense (>3 times/week)	47	6.29%
How often do you think you are on a healthy/balanced diet?		
Never	55	7.32%
Rarely	118	15.71%
Sometimes	203	27.03%
Frequently	343	45.67%
Always	32	4.26%
Chronic diseases		
Cardiovascular disease	27	3.60%
Diabetes mellitus	25	3.33%
High cholesterol	47	6.26%
High blood pressure	47	6.26%
Gastric disorders	41	5.46%
Intestinal disorders	21	2.80%
Obesity	50	6.66%
Other chronic diseases	32	4.26%

^(1)^ N = number of participants.

**Table 2 foods-10-00487-t002:** **The** relationship among BMI, age, and studied parameters.

Parameter (N ^(1)^ = 751)	Age (years) Mean ± SD Min, Max	BMI ^(2)^ (kg/m^2^) Mean ± SD Min, Max	*P* Value ^(3)^
Age (years)	38.02 ± 13.42 18, 80	24.60 ± 4.34 15.05, 43.57	*p <* 0.0001 ^(4)^ r ^(5)^ = 0.3948
18–29	23.18 ± 3.27	22.58 ± 3.67 16.13, 34.47	
30–39	34.38 ± 2.79	24.15 ± 3.93 15.05, 40.81	
40–49	44.28 ± 2.86	25.85 ± 4.38 17.82, 43.57	
50–59	52.78 ±2.71	26.75 ± 4.38 19.19, 40.40	
≥ 60	65.38 ± 4.75	25.88 ± 3.60 18.36, 38.6	
Gender			
Female	36.60 ± 12.80 18, 80	23.97 ± 4.43 15.05, 43.57	*p* < 0.0001 ^(6)^ r ^(7)^ = 0.4223
Male	41.04 ± 14.21 18, 80	25.91 ± 3.85 16.48, 40.81	*p* = 0.0061 ^(6)^ r ^(7)^ = 0.1765
General school	22 ± 3.46 18, 24	19.24 ± 0.89 18.28, 20.07	NA ^(8)^
High school	29.83 ± 12.71 18, 69	23.80 ± 3.90 17.14, 39.06	t(746) = 2.792 *p* = 0.0054 ^(9)^
College	40.46 ± 12.65 21, 80	24.85 ± 4.44 15.05, 43.57	
Environment			H(3) = 3.503 *p* = 0.1735 ^(10)^
Urban	38.49 ± 12.94 18, 80	24.65 ± 4.34 15.05, 43.57	
Suburban	38.61 ± 16.31 18, 77	25.36 ± 4.35 18.42, 35.15	
Rural	35.00 ± 15.16 18, 69	24.00 ± 4.34 17.00, 40.40	
Marital status			H(4) = 39.360 *p* < 0.0001 ^(10)^
Single	26.72 ± 9.79 18, 65	23.18 ± 4.05 15.05, 37.20	
Married/living together	41.44 ± 12.01 18, 80	25.21 ± 4.39 16.13, 43.57	
Divorced/separated	48.86 ± 7.81 27, 69	24.57 ± 4.06 18.06, 39.97	
Widow	58.20 ± 6.76 42, 64	24.93 ± 2.31 22.32, 30.29	
Employee status			F(2, 599) = 0.026 *p* = 0.9734 ^(11)^
Employed	41.39 ± 10.66 19, 77	25.19 ± 4.38 15.05, 43.57	
Unemployed	33.41 ± 10.11 21, 55	25.07 ± 4.62 16.40, 35.23	
Retired	64.20 ± 7.46 50, 80	25.28 ± 3.37 18.36, 35, 41	
Working student	21.61 ± 3.84 18, 47	22.19 ± 3.44 17.10, 33.56	
Professional area			H(7) = 25.67 *p* = 0.0003 ^(10)^
Nutrition	30.14 ± 11.36 19, 55	23.23 ± 3.72 17.09, 37.20	
Food	34.16 ± 13.44 19, 77	24.23 ± 5.15 17.04, 35.23	
Agriculture	37.77 ± 17.81 20, 77	28.62 ± 6.13 18.36, 39.06	
Sports	30.18 ± 9.93 18, 52	23.74 ± 2.67 18.33, 29.45	
Psychology	35.80 ± 14.27 19, 69	23.22 ± 3.91 17.10, 37.63	
Health-related activities	40.58 ± 12.68 18, 80	24.97 ± 4.14 17.00, 43.57	
Professional activity is not related to any of the above areas	38.23 ± 13.66 18, 74	24.61 ± 4.54 15.05, 40.81	
You are responsible for what you eat			U = 16,780 *p* = 0.3430 ^(12)^
Yes	37.97 ± 13.36 18, 80	24.56 ± 4.36 15.05, 43.57	
No	38.63 ± 14.34 18, 66	25.01 ± 4.07 17.10, 35.23	
Physical activity			H(4) = 6.0958 *p* = 0.1921 ^(10)^
Never	39.00 ± 11.33 21, 63	23.90 ± 4.69 16.40, 40.81	
Sporadic (<1 time/week)	40.57 ± 12.89 18, 69	25.10 ± 4.65 16.13, 43.57	
Occasionally (1 time/week)	36.35 ± 13.92 18, 80	24.57 ± 4.42 15.05, 39.06	
Moderate (2–3 times/week)	36.75 ± 13.50 18, 74	24.11 ± 3.79 17.00, 37.24	
Intense (>3 times/week)	35.93 ± 13.92 19, 80	24.39 ± 3.51 18.33, 32.43	
Hours/day spent watching TV or in front of the computer	4.81 ± 3.17 2, 20		*p* < 0.0001 ^(4)^ *r^x^* ^(5)^ = 0.1540 *p* = 0.001 ^(4)^ *r^y^* ^(5)^ = 0.1202
How often do you think you are on a healthy/balanced diet?			H(4) = 19.166 *p* = 0.0007 ^(10)^
Never	37.47 ± 12.14 18, 58	25.03 ± 4.34 16.90, 40.40	
Rarely	37.91 ± 13.18 18, 77	25.77 ± 5.48 16.40, 43.57	
Sometimes	35.20 ± 12.18 18, 65	25.04 ± 4.36 17.04, 39.06	
Frequently	39.81 ± 13.85 18, 74	24.07 ± 3.79 16.13, 37.20	
Always	38.03 ± 16.58 19, 80	22.26 ± 3.46 15.05, 29.21	
Chronic disease			N.P. ^(12)^
Cardiovascular disease	57.25 ± 12.15 32, 80	26.12 ± 4.53 18.36, 37.24	
Diabetes mellitus	45.40 ± 13.51 20, 64	28.54 ± 5.46 20.76, 40.40	
High cholesterol	51.72 ± 12.26 24, 77	27.17 ± 4.57 20.06, 37.63	
High blood pressure	48.82 ± 15.29 19, 80	28.52 ± 5.43 19.19, 43.57	
Gastric disorders	38.00 ± 13.68 19, 64	24.97 ± 5.02 15.05, 35.05	
Intestinal disorders	39.71 ± 13.65 19, 61	23.90 ± 3.97 16.48, 31.23	
Obesity	43.60 ± 12.91 19, 77	32.30 ± 4.17 29.74, 43.57	
Other chronic diseases	41.43 ± 11.31 19, 65	24.87 ± 4.24 15.05, 35.23	

^(1)^ N = number of participants; ^(2)^ BMI = body mass index; ^(3)^
*p* < 0.05 is considered significant; ^(4)^ Spearman test; ^(5)^ Spearman correlation; ^(6)^ Pearson test; ^(7)^ r = Pearson correlation; ^(8)^ NA = not applicable; this group was excluded from the analysis due to low number of cases; ^(9)^
*T*-test; ^(10)^ Kruskal–Wallis test; ^(11)^ ANOVA test; ^(12)^ Mann–Whitney test; r^x^ = correlation between hours/day spent watching TV or in front of the computer and age; r^y^ = correlation between hours/day spent watching TV or in front of the computer and BMI; ^(12)^ N.P. = not performed, because it is not relevant for the current research to compare BMI according to different pathologies (obesity is included, and there are subjects with more than one of the conditions asked).

**Table 3 foods-10-00487-t003:** Results of options regarding the motivations for health.

Question	1 n (%)	2 n (%)	3 n (%)	4 n (%)	5 n (%)
Q1	6 (0.80%)	20 (2.66%)	123 (16.38%)	300 (39.95%)	302 (40.21%)
Q2	28 (3.73%)	125 (16.64%)	274 (36.48%)	188 (25.03%)	136 (18.11%)
Q3	9 (1.20%)	49 (6.52%)	174 (23.17%)	333 (44.34%)	186 (24.77%)
Q4	8 (1.07%)	22 (2.93%)	160 (21.30%)	290 (38.62%)	271 (36.09%)
Q5	53 (7.06%)	125 (16.64%)	186 (24.77%)	257 (34.22%)	130 (17.31%)
Q6	16 (2.13%)	46 (6.13%)	181 (24.10%)	361 (48.07%)	147 (19.57%)
Q7	22 (2.93%)	147 (19.57)	212 (28.23%)	251 (33.42%)	119 (15.85%)
Q8	6 (0.80%)	19 (2.53%)	100 (13.32%)	286 (38.08%)	340 (45.27%)
Q9	54 (7.19%)	188 (25.03%)	197 (26.23%)	259 (34.49%)	53 (7.06%)
Q10	45 (5.99%)	65 (8.66%)	188 (25.03%)	189 (25.17%)	264 (35.15%)

1—totally disagree, 2—disagree, 3—neither agree nor disagree, 4—agree, 5—totally agree.

**Table 4 foods-10-00487-t004:** Item–item correlations for the composite scale investigating food properties ^(1)^.

Item	Q1	Q6	Q10
Q1	1.000		
Q6	0.434 **	1.000	
Q10	0.410 **	0.430 **	1.000

^(1)^ Cronbach alpha = 0.689, ** Correlation is significant at the 0.01 level (2-tailed).

**Table 5 foods-10-00487-t005:** Item–item correlations for the composite scale investigating health status and motivations ^(1)^.

Item	Q2	Q4	Q5R ^(2)^	Q7	Q8	Q9R ^(3)^
Q2	1.000					
Q4	0.357 **	1.000				
Q5R^(2)^	−0.251 **	−0.182 **	1.000			
Q7	−0.021	0.230 **	0.255 **	1.000		
Q8	0.341 **	0.656 **	−0.133 **	0.320 **	1.000	
Q9R^(3)^	0.150 **	0.108 **	0.259 **	0.040	0.144 **	1.000

^(1)^ Cronbach alpha = 0.517; ^(2)^ Q5R = reversed Q5; ^(3)^ O9R = reversed Q9, ** Correlation is significant at the 0.01 level (2-tailed).

**Table 6 foods-10-00487-t006:** Associations between food properties, health attitudes, and reported dietary behavior related to the investigated variables (*p*-value ^(1)^).

Parameter	Food Properties	Health Attitudes	Dietary Behavior
Age ^(2)^	X^2^(16, N = 751) = 26.03 *p* = 0.0535	X^2^(16, N = 751) = 26.60 *p* = 0.0353	X^2^(16, N = 751) = 28.57 *p* = 0.0270
Gender ^(2)^	X^2^(4, N = 751) = 2.8726 *p* = 0.5794	X^2^(4, N = 751) = 11.07 *p* = 0.0258	X^2^(4, N = 751) = 13.47 *p* = 0.0092
Environment ^(2)^	X^2^(8, N = 751) = 13.05 *p* = 0.1099	X^2^(8, N = 751) = 13.08 *p* = 0.109	X^2^(8, N = 751) = 15.53 *p* = 0.0496
Education level ^(2)^	X^2^ (8, N = 751) = 4.42 *p* = 0.8174	X^2^(8, N = 751) = 16.18 *p* = 0.0399	X^2^(8, N = 751) = 33.89 *p* = 0.0000
BMI ^(3)^	r = 0.0261 *p* = 0.4738	r = 0.0683 *p* = 0.0612	r = −0.0038 *p* = 0.292
Physical activity ^(2)^	X^2^ (16, N = 751) = 43.27 *p* = 0.0003	X^2^ (16, N = 751) = 33.13 *p* = 0.0071	X^2^ (16, N = 751) = 52.06 *p* = 0.0000
Hours/day watching TV/PC ^(3)^	r = −0.0337 *p* = 0.3551	r = 0.0901 *p* = 0.0134	r = 0.0230 *p* = 0.5291
Cardiovascular disease ^(2)^	X^2^ (4, N = 751) = 4.14 *p* = 0.3869	X^2^ (4, N = 751) = 15.01 *p* = *0.0047*	X2(4, N = 751) = 6.84 *p* = 0.1445
Diabetes mellitus ^(2)^	X^2^ (4, N = 751) = 2.40 *p* = 0.6618	X^2^ (4, N = 751) = 2.55 *p* = 0.6345	X^2^ (4, N = 751) = 0.86 *p* = 0.9300
High cholesterol ^(2)^	X^2^ (4, N = 751) = 0.67 *p* = 0.9546	X^2^ (4, N = 751) = 4.59 *p* = 0.3315	X^2^ (4, N = 751) = 1.37 *p* = 0.8488
High blood pressure ^(2)^	X^2^ (4, N = 751) = 5.56 *p* = 0.2340	X^2^ (4, N = 751) = 3.86 *p* = 0.4240	X^2^ (4, N = 751) = 2.96 *p* = 0.5632
Gastric disorders ^(2)^	X^2^ (4, N = 751) = 13.85 *p* = 0.0078	X^2^ (4, N = 751) = 11.85 *p* = 0.0185	X^2^ (4, N = 751) = 16.15 *p* = 0.0028
Intestinal disorders ^(2)^	X^2^ (4, N = 751) = 9.95 *p* = 0.0411	X^2^ (4, N = 751) = 6.98 *p* = 0.1366	X^2^ (4, N = 751) = 8.01 *p* = 0.0911
Obesity ^(2)^	X^2^ (4, N = 751) = 3.09 *p* = 0.5416	X^2^ (4, N = 751) = 4.45 *p* = 0.3485	X^2^ (4, N = 751) = 26.47 *p* = 0.0000
Other ^(2)^	X^2^ (4, N = 751) = 3.86 *p* = 0.4240	X^2^ (4, N = 751) = 6.90 *p* = 0.4100	X^2^ (4, N = 751) = 10.85 *p* = 0.8863

^(1)^*p* < 0.05 was considered significant; ^(2)^ Chi square test for n x m table; ^(3)^ Spearman test.

## Data Availability

Data are available from the corresponding author upon reasonable request.

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
