# Peer review of "Demographic, Anthropometric and Food Behavior Data towards Healthy Eating in Romania"

_foods, 2021, doi:10.3390/foods10030487_

Round 1

Reviewer 1 Report

This is a good quality paper. The scientific objective is clear, the methodology is correct and the results are presented in the right way.

In Table 1, hours/day spent watching TV or in front of the computer are not presented appropriately.

In Table 5, there are some Χ2 and some Χ2. Please make it consistent.

page 2, line 6 - established

2, 9 - consumer behaviour

2, 11 from the bottom - as the first

4, 9 - using the

13, 2 - study,

13, 9 - study,

14, 1 - [56],

14, 2 - baseline,

14, 4 - logical

14, 20 - was significantly

14, 22 - disorders,

Reviewer 2 Report

Thank you for making modifications based on my previous suggestions.

I feel there are a few ways you could still improve this paper. 

  1. The relationship between demographics and BMI - this is reported clearly
  2. The relationship between demographics and perceptions about food properties (C1)
  3. The relationship between demographics and health attitudes (C2)
  4. The relationship between demographics and reported dietary behaviour (Q3)

Regarding 2. 3. and 4. it would be easier for the reader if you referred to these by a name (e.g. food properties) in the tables and text, rather than a number C1.

Regarding Table 5 - you have stated significant differences using Chi squared tests, which is a group difference test. However, you give no indication of where the differences lie, or of any patterns. For example if there are age differences, where are they given there are five age groups? Is it just the youngest group is different from the rest? or the older group? Are all of the groups different from each other. This is a significant limitation of you analysis and limits what you can say and recommend from the findings. For example, if concern for food properties declined with age, perhaps intervention strategies could be created to address this.

Is there a relationship between reported behaviour and health attitudes? Theoretically there should be - people who have high health motivations, in the absence of external barriers, should have more healthy dietary behaviour. This may differ across demographic characteristics as well. This represents a significant opportunity in your data.

I would like to see more discussion of the implications of these findings for researchers and practitioners. What should be done to improve the situation? These recommendations should arise from your findings. You did not measure knowledge, so recommending actions to increase knowledge are not well founded.

The language still needs significant improvement. For example, "taking over the American model" sounds like the changes in Romania are larger than the changes in America. Whereas I think you mean it is "taking after the American model". Also in discussion "Since we already proved that the age at which BMI starts to influence glycaemia..." proved is a very strong word scientifically, and you are citing a pilot study. I am not an expert in physiology, but I understand glycaemia and BMI may be associated, but which one causes which? A better phrase would be "Since we have already shown a significant positive association between BMI and glycaemia after the age of 22..."

These are just two examples, there are many more language errors and inconsistencies. Professional English editing services may be warranted.   

Round 2

Reviewer 2 Report

Thank you for addressing my comments in the last revision.

Author Response

Dear Reviewer,

Thank you for evaluating the article entitled “Demographic, Anthropometric and Food Behavior Data towards Healthy Eating in Romania”, authored by Anca Bacârea, Vladimir Constantin Bacârea, Cristina Cînpeanu, Claudiu Teodorescu, Ana Gabriela Seni, Raquel Guiné, Monica Tarcea.

We have tried to answer your requirements, as follows:

  • We improved the introduction;
  • We improved the conclusions;
  • We made English correction of the article, and we believe that the results are much clearer now.

Thank you for your recommendations,

Kind regards,

The authors

This manuscript is a resubmission of an earlier submission. The following is a list of the peer review reports and author responses from that submission.

Round 1

Reviewer 1 Report

Thank you for the opportunity to review your paper "Demographic, Anthropometric and Food Behavior Data towards Healthy Eating in Romania". It is important to understand which factors are linked to healthy eating behaviors, particularly in understudied countries, or those countries that have experienced significant societal change in recent times. I congratulate you for pursuing this in research agenda in Romania. 

I have a number of concerns with the manuscript in its current form, concerning your analyses and results. I feel you should rework these sections, which may provide more insights for you to discuss in the Discussion.  

I would suggest splitting Table 1. First a general demographics table, to give an overview of your sample, with each of the parameters in Table 1, but not split by age and BMI (the 3rd and 4th columns). As it is, I can't tell how many in your sample are in meaningful BMI groups (e.g. underweight, normal weight, and overweight/obese). If you didn't have these three groups, giving the mean BMI for the entire sample is useful.

Then a second table could explore the relationships between BMI, age and other parameters. If age was collected as a continuous variable, I would analyse it that way (it is not clear in the methods if stratification occurs after data collection - or if participants nominated their age category). All statistical tests should be presented with the parameter estimate, and significance level (never as just the significance level as shown in Column 5 of Table 1 and in Table 5). For example, a t-test should be t(48)=2.3, p=0.026; and a Chi test should be X2(1, N=104)=1.7, p=0.04. 

The 10 questions (pg 3, lines 108-119) are a mixture of questions about food composition (Q1, Q6, Q10) and Health (Q2, Q3, Q4, Q5, Q7, Q8, Q9). Q3 is a "frequency" based question, whereas many others are attitudinal (a positive or negative affect towards the subject of the question). Q5 & Q9 are reversed - the 'healthier' position is on the left (disagree) - compared to the others where the 'healthier' position is on the right (agree).

Given these items are measuring different things - and Cronbach’s alpha is a measure of internal consistency, (that is, how closely related a set of questions are as a group) Table 4 does not make sense, neither does the global alpha at line 182. It is not clear what you are measuring to determine an alpha for each single question (given Cronbach's alpha is used on groups of questions) and you don't go on to do anything with that. A related point - what is the purpose of Table 3? It you are attempting to show that questions are correlated - and therefore could be used as a composite scale (for health motivation, for example) you would need to show the significance of these correlations.

If is suited your purpose - you could attempt the creation of a composite scale for health attitudes/motivations (Q2, Q4, Q5, Q7, Q8, Q9) - ensuring Q5 & Q9 are reversed first. Then a composite scale for 'food properties' (Q1, Q6, Q10). That leaves Q3 standalone (healthy diet frequency). Use correlations and Cronbach's alpha to demonstrate the validity of combining these items into a composite. And then explore the relationships between the parameters (column 1 of Table 5) and these three composites. By approaching it this way you could discuss the relationships in terms of three dimensions (health, diet frequency and food properties) rather than for each question - which currently is a bit meaningless. If you go this way - ensure you justify why it is logical to investigate these three dimensions, AND why the questions can be grouped into composites this way. You would need to discuss amongst yourselves what the dimensions could be and why each question fits under each dimension (you might go with more or less than three dimensions), then you could say you based your decision on face validity of the questions as determined by the researchers, and decide whether to supplement this justification with other measures.

Once you have reworked the methods and results - you can then strengthen the introduction to justify why you would explore what you have and briefly outline the literature in the area.

Then discuss your findings comprehensively in light of current literature. At present the discussion is too much re-stating the results (and including numbers and statistics in the discussion which should appear in the results).

I wish you well with future revisions.   

Reviewer 2 Report

This is a good quality paper. The research question is relevant and important. The methodology is correct, and the results are clearly presented. It significantly advances the state of the art in understanding demographic and anthropometric determinants of diet healthiness on the example of a Romanian sample.

lines 24, 101, 158, 307, page 6 - spent

line 244 - What does "the payment" mean?

Reviewer 3 Report

The aim of this study was to evaluate demographic data, anthropometric data and elements related to food behavior, as well as the Romanian people motivations towards healthy eating, as first step for further development of health policies and strategies to improve eating behavior.

The authors observe that in most cases the healthy subjects are more preoccupied with a healthy diet, excepting those with cardiovascular disorders, who are more preoccupied with low fat diet, diet rich in vitamins and minerals and avoidance of processed foods.

The paper is well written. The introduction, methos and discussion are well presented.

The tables are confused. Please improve it. In addiction, in table 1 please add the min,max also in all age's subgroups.